# Clinical Utility and Validation of the Krakow DCM Risk Score—A Prognostic Model Dedicated to Dilated Cardiomyopathy

**DOI:** 10.3390/jpm12020236

**Published:** 2022-02-08

**Authors:** Ewa Dziewięcka, Mateusz Winiarczyk, Sylwia Wiśniowska-Śmiałek, Aleksandra Karabinowska-Małocha, Matylda Gliniak, Jan Robak, Monika Kaciczak, Przemysław Leszek, Małgorzata Celińska-Spodar, Marcin Dziewięcki, Paweł Rubiś

**Affiliations:** 1Department of Cardiac and Vascular Diseases, Jagiellonian University Collegium Medicum, John Paul II Hospital, 31-202 Krakow, Poland; swisniowskasmialek@gmail.com (S.W.-Ś); akarabinowska@gmail.com (A.K.-M.); 2Students’ Scientific Group at Department of Cardiac and Vascular Diseases, Jagiellonian University Collegium Medicum, John Paul II Hospital, 31-008 Krakow, Poland; winiarczyk.mateusz@gmail.com (M.W.); glimat@interia.pl (M.G.); jahu114@gmail.com (J.R.); monikakaciczak@gmail.com (M.K.); 3Department of Cardiovascular Surgery and Transplantology, Jagiellonian University Collegium Medicum, John Paul II Hospital, 31-008 Krakow, Poland; 4Department of Heart Failure and Transplantation, The Cardinal Stefan Wyszyński Institute of Cardiology, 04-628 Warsaw, Poland; pleszek@ikard.pl; 5Department of Anaesthesiology and Intensive Care, The National Institute of Cardiology, 04-628 Warsaw, Poland; mcelinska@ikard.pl; 6College of Economics and Computer Science (WSEI), 31-150 Krakow, Poland; marcin.dziewiecki@gmail.com

**Keywords:** dilated cardiomyopathy, non-ischemic heart failure with reduced ejection fraction, prognosis, prognostic model, mortality risk, Krakow DCM Risk Score

## Abstract

Background: One of the most common causes of heart failure is dilated cardiomyopathy (DCM). In DCM, the mortality risk is high and reaches approximately 20% in 5 years. A patient’s prognosis should be established for appropriate HF management. However, so far, no validated tools have been available for the DCM population. Methods: The study population consisted of 735 DCM patients: 406 from the derivation cohort (previously described) and 329 from the validation cohort (from 2009 to 2020, with outcome data after a mean of 42 months). For each DCM patient, the individual mortality risk was calculated based on the Krakow DCM Risk Score. Results: During follow-up, 49 (15%) patients of the validation cohort died. They had shown significantly higher calculated 1-to-5-year mortality risks. The Krakow DCM Risk Score yielded good discrimination in terms of overall mortality risk, with an AUC of 0.704–0.765. Based on a 2-year mortality risk, patients were divided into non-high (≤6%) and high (>6%) mortality risk groups. The observed mortality rates were 8.3% (*n* = 44) vs. 42.6% (*n* = 75), respectively (HR 3.37; 95%CI 1.88–6.05; *p* < 0.0001). Conclusions: The Krakow DCM Risk Score was found to have good predictive accuracy. The 2-year mortality risk > 6% has good discrimination for the identification of high-risk patients and can be applied in everyday practice.

## 1. Introduction

Dilated cardiomyopathy (DCM) is the commonest indication for heart transplantation and the third most common cause of heart failure (HF) [1,2,3,4,5]. It is characterized by left ventricular (LV) systolic dysfunction and LV enlargement in the absence of significant coronary artery disease and abnormal loading conditions [1,2,3,4,5]. Over the past few decades, the aetiology and natural history of DCM have been thoroughly elucidated, demonstrating that various aetiologies causing LV dysfunction may manifest with the same clinical phenotype as DCM.

Providing accurate prognoses in HF can pose numerous challenges. So far, many scales, including BCN Bio-HF, CHARM, EMPHASIS, HFSS, MAGGIC, MUSIC and SHFM, all dedicated to the general HF cohort, have been developed, with a diagnostic accuracy of between 60% and 80% [6,7,8,9,10,11,12,13,14,15]. However, most were created 10–20 years ago, before the global implementation of HF modifying therapies. These treatments substantially diminish the applicability of these scales to current HF populations, especially given the fact that they have not been validated for any subgroup of HF, including DCM. The validation of the scales is of utmost importance to DCM patients, as they differ substantially from other types of HF patients in terms of their younger age and their presentation of fewer comorbidities, which leads to an overall lower mortality rate [3,16].

At present, there are only two prognostic scales dedicated to DCM: Miura et al. and the Krakow DCM Risk Score [12,17,18]. Although the first is a simple numerical score based on five parameters (and as such is easy to calculate), its prognostic value is questionable, especially since it was developed before current HF therapies were introduced [18]. The second one is a linear scale that has performed very well in bootstrapping, and despite its complexity, an online tool is now available. Nevertheless, it has not as yet been externally validated [17,18].

Therefore, the aim of this work is to externally validate the Krakow DCM Risk Score and to establish a cut-off point for high-risk DCM patients.

## 2. Materials and Methods

### 2.1. Patient Population

The derivation cohort comprised of 406 DCM patients (aged 54 ± 14 years, 81% male, NYHA class 2.5 ± 0.9, LVEF 26 ± 9%, left ventricle end-diastolic diameter—66 ± 10 mm, mean NT-proBNP 1476 pg/mL) [18]. The validation cohort consisted of 329 consecutive DCM patients from 7 polish cardiac centres from 2009 to 2020 with complete baseline data, 118 (35%) from 2009 to 2015, and 215 (65%) from 2016 to 2020; the distinction is motivated by the publication of the European Society of Cardiology (ESC) HF guidelines in 2016. Patients underwent detailed diagnostic work-up (clinical evaluation, laboratory tests, electrocardiogram—ECG, echocardiography and invasive coronary angiography or computed tomography coronary angiography, as shown in Table 1) [1,19,20,21]. DCM was diagnosed following the previously published ESC criteria, based on (1) the presence of LV dilation and impaired systolic function (LV ejection fraction—LVEF < 45%) detected via echocardiography and (2) the exclusion of significant coronary artery disease, primary heart valve disease, congenital heart disease and severe arterial hypertension [1,3,21,22].

### 2.2. Clinical Follow-Up and Endpoint Definition

As in the original derivation study, the endpoint was all-cause mortality. Between May and September 2021, information on the status of the patients was collected through publicly available databases, medical records and telephone contact.

### 2.3. Structure of the Krakow DCM Risk Score

The Krakow DCM Risk Score is a multivariable linear risk model that predicts all-cause mortality in DCM patients [17,19]. It consists of sex, age, symptoms duration and severity, comorbidities (diabetes mellitus, stroke, liver and kidney diseases, dyslipidaemia, anaemia), LBBB, LV size and systolic function, NT-proBNP and HF therapy implementation. For all DCM patients, 1-, 2-, 3-, 4- and 5-year mortality risks were calculated according to the previously published formula (Appendix B) [19].

### 2.4. Statistical Analysis

All parameters are presented as mean ± standard deviation (SD) or counts and percentages. The continuous parameters were tested for their normal distribution with the Shapiro–Wilk test. Comparisons of quantitative variables were conducted with t-tests or the Mann–Whitney test for data with and without normal distribution; the Chi-square test was performed in the case of qualitative parameters. Areas under the receiver operating curve (AUC) were calculated to assess the accuracy of the Krakow DCM Risk Score for the prediction of 1-, 2-, 3-, 4- and 5-year mortality. Kaplan–Meier analyses were performed for the calculation of observed mortality and the log-rank test for the comparisons of mortality rates. Results were considered statistically significant when their *p*-value was <0.05. The Statistica package, version 13.0 (StatSoft, TIBCO Software Inc., Palo Alto, CA, USA), was used for the statistical analysis.

## 3. Results

### 3.1. Baseline Characteristics

During a follow-up of mean 41.6 ± 29.3 months, 49 (15%) patients of the validation cohort died: 41 (85%) stemming from cardiovascular causes (38 patients—due to HF worsening, 3 patients—SCD), and eight patients died as a result of neoplasms. In terms of procedures, 12 patients underwent left ventricle assistant device (LVAD) implantations and six patients heart transplants (HTX); one patient received both procedures.

Deceased were older, more symptomatic, had more severe LV and right ventricular (RV) remodelling with worse LV and RV systolic function, more often had significant tricuspid regurgitation, anaemia and chronic kidney disease, higher levels of N-terminal pro-b-type natriuretic peptide (NT-proBNP), and they required higher loop diuretics dosages (Table 1). HF modifying therapies, such as beta blockers and renin–angiotensin–aldosterone system inhibitors, were more commonly used in both groups, whereas digoxin, higher doses of loop diuretics and CRT were more prevalent among deceased.

### 3.2. Performance of Krakow DCM Risk Score

Calculated mortality risks, based on the Krakow DCM Risk Score, significantly differed between alive and deceased patients (Table 2). The model under analysis yielded good discrimination in terms of overall 1-, 2-, 3-, 4- and 5-year mortality with an AUC of 0.704–0.765 (Figure 1).

The validation cohort differed significantly from the derivation cohort in terms of age, HF duration and symptoms, comorbidities (obesity—body mass index, dyslipidaemia, anaemia and chronic kidney disease), heart rate, NT-proBNP and their required loop diuretics dosage (Appendix A, Appendix C). However, they did not differ in terms of echocardiographic findings and mortality rates (*p* = 0.97) (Appendix A).

### 3.3. High Mortality Risk DCM Patients

Although the mean observation period for the whole DCM cohort (derivation and validation cohorts) was 45 months, only 60% (*n* = 425) of the entire population had follow-up longer than 3 years, 50% (*n* = 356) longer than 4 years and 39% (*n* = 273) longer than 5 years. Therefore, high mortality risk was assessed based on 2-year mortality risk as calculated by the Krakow DCM Risk Score in those patients (*n* = 735) with available 2-year follow-up data.

Thus, 735 DCM patients were divided into high and non-high mortality risk groups on the basis of 1st–3rd quartile vs. 4th quartile as calculated by the Krakow DCM Risk Score (≤6.0% vs. >6.0%). The observed mortality rates were 8.3% (*n* = 44) vs. 42.6% (*n* = 75), respectively (HR 3.37 (95%CI 1.88–6.05), *p* < 0.0001) (Figure 2, Appendix A). The cut-off point of 6.0% had a high prognostic accuracy of 0.77.

## 4. Discussion

The study findings can be summarised as follows: the Krakow DCM Risk Score yielded adequate discrimination in terms of overall mortality in the DCM population. The cut-off point of 6% for a 2-year mortality risk displayed good discrimination for high mortality risk DCM patients. 

### 4.1. Prognostic Models in DCM

The observed mortality in DCM patients is high, and over the course of 5 years reaches approximately 20% but varies among different studies that have been carried out [12,19,23,24,25,26]. However, due to its unique features (such as its occurrence at a young age, LV reverse remodelling and fewer comorbidities), risk stratification in DCM cannot be accurately performed using unspecific models developed for broad HF populations [27,28].

Although numerous prognostic parameters have been established in DCM, including HF symptoms severity (mostly assessed by semi-quantitative NYHA class), LV and RV systolic function and size, comorbidities (e.g., diabetes mellitus, anaemia, chronic kidney disease), cardiac fibrosis, or ventricular arrhythmias, heir clinical meaning in isolation has limited value for more thorough-going risk stratification [12,28,29,30,31,32]. Consequently, until recently, there were no tools in existence for accurate mortality risk stratification in DCM. So far, two prognostic models dedicated to DCM patients exist: (1) the Miura et al. score and (2) the Krakow DCM Risk Score [12,19]. Miura et al. is a numerical score based on a Japanese national DCM survey from the 1990s, which calculates the 5-year mortality risk based on just five parameters: sex, age, NYHA class, LV diameter and LVEF [12]. Although the calculation is straightforward, its overall performance is far from satisfactory [19]. Moreover, apart from our own external validation in a contemporary European cohort, the Miura score has never been validated. On the other hand, the Krakow DCM Risk Score, based on 406 DCM patients from 2010 to 2019, is a linear model that allows for the calculation of the individual mortality risk at any given time, preferably between 1 and 5 years [17,19]. To facilitate the use of the proposed model, an online calculator has been created, which is available on the Heart Failure Association of the Polish Cardiac Society official webpage (Figure 3).

### 4.2. Krakow DCM Risk Score Performance

According to the analysis presented here, the Krakow DCM Risk Score shows adequate performance in external validation, with an accuracy of over 70% [14]. This precision is comparable to the most widely available tools currently in use, such as the GRACE risk score 2.0 for mortality outcomes in acute cardiac syndrome, the HCM Risk-SCD score for sudden cardiac death (SCD) in hypertrophic cardiomyopathy (HCM), or CHA_2_DS_2_-VASc for stroke in atrial fibrillation, and is at least similar to prognostic scores in general HF cohorts, including the Heart Failure Survival Score (HFSS), the Seattle Heart Failure Model (SHFM) and the Meta-Analysis Global Group in Chronic Heart Failure (MUSIC) [14,15,33,34,35,36]. The Krakow DCM Risk Score provided good discrimination for at least 7 out of 10 patients. Taking into account the significant differences between the derivation and validation cohorts, including age, HF symptoms and NT-proBNP level, their outcomes were poor with similar 5-year mortality of over 20%. Moreover, the high accuracy of the Krakow DCM Risk Score, despite the diversity of the DCM cohort’s understudy, strengthens its value in various DCM cohorts.

### 4.3. Identification of High Mortality DCM Patients

Accurate prediction of long-term outcomes, including mortality, is a cornerstone of comprehensive HF management. Although the overall prognosis in DCM is poor, patients’ individual prognoses may be highly variable. Therefore, the development of an accurate prognostic risk model has the potential for more comprehensive and tailored management. Quantifying patients’ survival predictions based on their overall risk profile can help identify those in need of more concentrated monitoring and more intensive HF therapy. Additional potential use of the model includes educating patients on the significant value of HF medications. When altering their implementation in the online calculator, patients can be presented with a higher mortality risk without proper pharmacotherapy. Moreover, patients with high mortality risk should be earlier listed for cardiac transplantation or counselled about end-of-life issues.

As stated above, the identification of patients with high mortality risk is crucial in everyday practice in HF and DCM patients. We recognized the high mortality risk as a 2-year mortality risk estimated above 6.0% (75th percentile). The accuracy of this cut-off point is high, and patients with a calculated risk of >6.0% had a five-times higher mortality risk during follow-up. Therefore, it can be used in everyday practice for HF therapy qualification, especially in terms of invasive procedures.

### 4.4. Limitations

Although the size of the study population is large, with over 700 DCM patients, taking into account DCM epidemiology, this is still a one-country retrospective analysis. The mean observation period for the whole DCM cohort was 45 months; however, only half of the population had follow-up data on 4-year mortality. Therefore, to make the model more accurate, high mortality risk was assessed based on observations over the course of 2 years. Only 19% of patients had been treated with angiotensin receptor-neprilysin inhibitors; however, sacubitril/valsartan was only available for the second half of the study. Moreover, the Krakow DCM Risk Score does not include HTX and LVAD implantation as outcomes; however, overall mortality is the definite endpoint. Although the Krakow DCM Risk Score has a complicated linear model, it includes the most known DCM prognostic parameters, and its discrimination is good. Therefore, to facilitate the use of the model, an online calculator has been created and made widely available.

## 5. Conclusions

The overall mortality risk in the DCM population is high and reaches 23% during 5-year follow-up. A DCM-dedicated prognostic model, namely the Krakow DCM Risk Score, was found to have good predictive accuracy. The 2-year mortality risk of over 6.0% has good discrimination for the identification of high-risk patients and can be used in everyday practice.

## Figures and Tables

**Figure 1 jpm-12-00236-f001:**
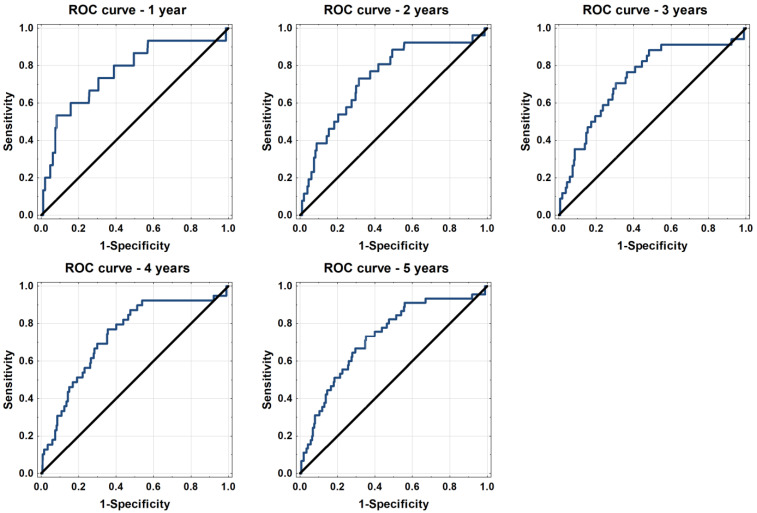
ROC curves for the performance of the Krakow DCM Risk Score.

**Figure 2 jpm-12-00236-f002:**
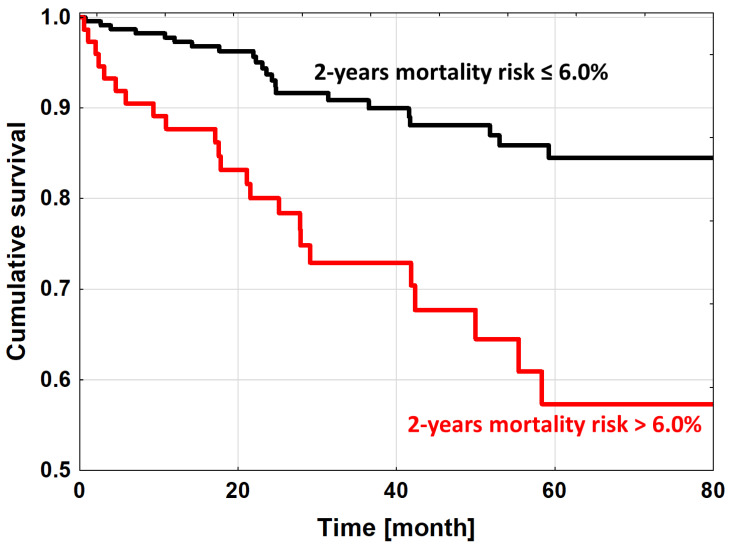
The Kaplan–Meier survival for high (4th quartile) and non-high (from 1st to 3rd quartiles) mortality risk groups divided according to calculations based on the Krakow DCM Risk Score.

**Figure 3 jpm-12-00236-f003:**
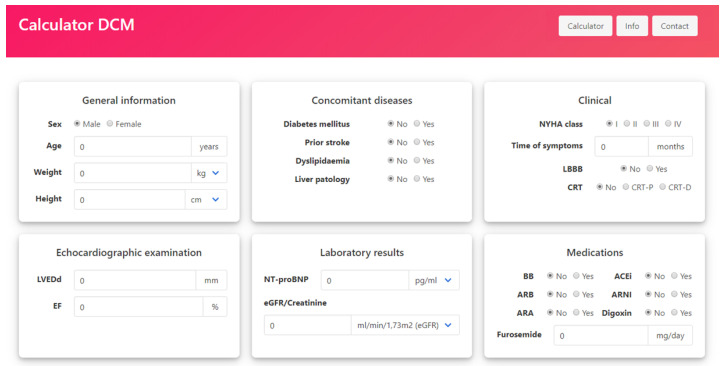
Online tool for the calculation of individual mortality risk based on the Krakow DCM Risk Score (available on the Heart Failure Association of the Polish Cardiac Society official webpage: https://www.niewydolnosc-serca.pl/kalkulator.html; accessed on 4 November 2021).

**Table 1 jpm-12-00236-t001:** Baseline characteristics.

Parameters	All *n* = 329	Alive *n* = 280 (85%)	Deceased *n* = 49 (15%)	*p*-Value
**Clinical parameters**				
Age (years)	49.88 ± 10.71	49.31 ± 10.69	53.29 ± 10.66	0.006
Male (*n* (%))	261 (79.3%)	226 (80.7%)	35 (71.4%)	0.14
Symptoms duration (months)	21.3 ± 34.78	25.15 ± 89.865	28.63 ± 41.03	0.31
BMI (kg/m^2^)	29.39 ± 14.37	29.47 ± 16.162	29.16 ± 6.18	0.77
NYHA class	2.23 ± 0.77	2.14 ± 0.73	2.73 ± 0.8	**<0.0001**
NYHA III/IV	89 (27.1%)	63 (22.5%)	26 (53.1%)	**<0.0001**
Diabetes mellitus (*n* (%))	64 (19.5%)	49 (17.5%)	15 (30.6%)	0.03
Prior stroke (*n* (%))	13 (4%)	11 (3.9%)	2 (4.1%)	0.96
Liver diseases (*n* (%))	47 (14.3%)	39 (13.9%)	8 (16.3%)	0.66
Dyslipidaemia (*n* (%))	258 (78.4%)	223 (79.6%)	35 (71.4%)	0.20
COPD (*n* (%))	18 (5.5%)	15 (5.4%)	3 (6.1%)	0.83
Atrial fibrillation (*n* (%))	100 (30.4%)	84 (30.0%)	16 (32.7%)	0.71
Hypertension (*n* (%))	155 (47.1%)	127 (45.4%)	28 (57.1%)	0.13
SBP (mmHg)	122.99 ± 20.55	123.97 ± 20.58	117.83 ± 19.94	0.09
**ECG findings**				
HR (bpm)	76.5 ± 16.62	75.76 ± 16.011	80.93 ± 19.75	0.07
QRS (ms)	105.59 ± 35.24	104.11 ± 34.077	114.49 ± 41.13	0.05
LBBB (*n* (%))	74 (22.5%)	62 (22.1%)	12 (24.5%)	0.72
VT (0/1)	88 (26.7%)	74 (26.6%)	14 (28.6%)	0.81
**Echocardiographic findings**				
LVEF (%)	27.02 ± 9.96	27.73 ± 9.936	23.72 ± 9.55	**0.01**
LVEDd (mm)	65.08 ± 8.89	64.76 ± 8.64	66.46 ± 10.24	0.24
IVS (mm)	10 ± 1.98	10.06 ± 1.987	9.7 ± 1.91	0.33
RVOT (mm)	33.79 ± 6.61	33.4 ± 6.383	36.15 ± 7.41	**0.008**
TAPSE (mm)	19.14 ± 4.12	19.41 ± 4.118	17.77 ± 4.02	**0.01**
LAA (cm^2^)	28.97 ± 8.33	28.35 ± 7.813	32.04 ± 10.27	**0.01**
RVSP (mmHg)	25.46 ± 13.12	23.89 ± 11.886	33.46 ± 16.15	**0.0001**
MR moderate/severe (*n* (%))	111 (33.7%)	89 (31.8%)	22 (44.9%)	0.07
TR moderate/severe (*n* (%))	66 (20.1%)	44 (15.7%)	22 (44.9%)	**<0.0001**
**Laboratory tests results**				
Hb (g/dL)	14.55 ± 1.66	14.67 ± 1.626	13.88 ± 1.78	**0.002**
eGFR (ml/min/1.73 m^2^)	83.5 ± 20.9	84.91 ± 20.411	75.67 ± 22.71	**0.006**
NT-proBNP (pg/mL)	2759.25 ± 3639.6	2297.9 ± 3131.6	4980.7 ± 4910.9	**<0.0001**
LDL (mmol/L)	2.99 ± 0.98	2.99 ± 0.969	2.86 ± 1.04	0.36
**Heart failure therapy**				
BB (*n* (%))	317 (96.4%)	272 (97.1%)	45 (91.8%)	0.049
ACEi/ARB/ARNI (*n* (%))	291 (88.4%)	253 (90.4%)	38 (77.6%)	**0.01**
MRA (*n* (%))	285 (86.6%)	244 (87.1%)	41 (83.7%)	0.51
Loop diuretics (mg/d) ^1^	44.47 ± 69.24	37.91 ± 56.42	80.16 ± 113.94	**0.0003**
Furosemide (mg/d)	25.7 ± 50.31	22.02 ± 44.717	45.57 ± 72	0.03
Ivabradine (*n* (%))	53 (16.1%)	41 (14.6%)	12 (24.5%)	0.08
Digoxin (*n* (%))	52 (15.8%)	38 (13.6%)	14 (28.6%)	**0.008**
Statins (*n* (%))	148 (45%)	124 (44.3%)	24 (49.0%)	0.54
CRT (*n* (%))	11 (3.3%)	6 (2.1%)	5 (10.2%)	**0.004**
ICD (*n* (%))	30 (9.1%)	23 (8.2%)	7 (14.3%)	0.17

^1^ Loop diuretics dosages were calculated as the sum of the daily furosemide dosage and 3 times the torsemide daily dosage. Abbreviations: BMI—body mass index; NYHA—New York Heart Association class; COPD—chronic obstructive pulmonary disease; SBP—systolic blood pressure; HR—heart rate; LBBB—left bundle branch block; VT—ventricular tachyarrhythmia; LVEF—left ventricle ejection fraction; LVEDd—left ventricle end-diastolic diameter, obtained from parasternal long-axis view (PLAX); IVS—intraventricular septum thickness obtained from PLAX; RVOT—right ventricle outflow track diameter obtained from PLAX; MR/TR—mitral/tricuspid regurgitation; Hb—haemoglobin; eGFR—estimated glomerular filtration rate; NT-proBNP—N-terminal prohormone B-type natriuretic peptide; LDL—low-density lipoprotein; BB—beta blocker; ACEI—angiotensin-converting enzyme inhibitor; ARB—angiotensin receptor blocker; ARNI—angiotensin receptor—neprilysin inhibitor; MRA—mineralocorticoid receptor antagonist; ICD—implantable cardioverter–defibrillator; CRT—cardiac resynchronization therapy.

**Table 2 jpm-12-00236-t002:** Krakow DCM Risk Score mortality rates observed and predicted at 1, 2, 3, 4 and 5 years.

Follow-Up	Observed Mortality [%] (Kaplan–Meier Analysis)	Calculated Mortality Risk [%]				Krakow DCM Risk Score Discrimination	
		All	Alive	Deceased	*p*-Value	AUC-ROC	*p*-Value
1 year	4.68 ± 0.02	3.52 ± 9.13	3.08 ± 8.41	11.2 ± 15.93	0.0006	0.765 [95%CI 0.628–0.902]	0.0001
2 years	9.96 ± 0.02	6.88 ± 14.21	5.96 ± 13.02	14.48 ± 20.16	0.0003	0.718 [95%CI 0.613–0.822]	<0.0001
3 years	14.41 ± 0.02	10.37 ± 18.68	8.25 ± 15.4	18.91 ± 25.4	0.0002	0.706 [95%CI 0.608–0.805]	<0.0001
4 years	17.60 ± 0.03	13.06 ± 21.32	10.32 ± 18.54	20.89 ± 26.19	0.0001	0.709 [95%CI 0.616–0.802]	<0.0001
5 years	22.57 ± 0.03	15.78 ± 23.53	11.51 ± 19.85	23.49 ± 27.26	0.0002	0.704 [95%CI 0.609–0.798]	<0.0001

## Data Availability

Data available on request due to privacy restrictions. The data presented in this study are available on request from the corresponding author.

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
