# Peer review of "Clinical Utility and Validation of the Krakow DCM Risk Score—A Prognostic Model Dedicated to Dilated Cardiomyopathy"

_jpm, 2022, doi:10.3390/jpm12020236_

Round 1

Reviewer 1 Report

Although there is a plethora of prognostic scales for prognostic assessment in heart failure, there’s no one validated for a subgroup of heart failure patients affected by dilated cardiomyopathy. This subgroup has epidemiological significative differences from others heart failure patients, so it’d valuable to research and validate a dedicated prognostic score. The aim of the authors was to validate the previously investigated Krakow DCM Risk Score. My specific comments are as follows:

Methods, population: despite is correct to cite the derivation cohort, should be useful report in this section a comparison with the validation cohort to find potential differences. Furthermore, it seems to be important an explanation of the centres participating in the validation study (only Polish hospitals?), in order to empower the study’s results.

Method, end point definition: all cause mortality as unique endpoint seems to be a little unambitious. Why didn’t the authors consider only CV mortality or any other – maybe secondaries – endpoints, such heart transplantation o LVAD implantation?

Results, baseline characteristics: these data refer to the derivation, the validation cohorts or a sum up of the two? By the numbers seem to refer to the validation cohort but is useful to declare it. It is also useful to explain more details about this population even before the subgroups statement, to better understand if the results are conditioned by selection bias. Furthermore, seems to be useful, to refine the cardiovascular prognostic power of the score, to exclude deaths caused by neoplasms. The term “non survivors” also isn’t accurate: a patient with transplanted heart is considered a survivor but its life was saved by proper clinical management or by less abrupt onset of refractory heart failure?

Results, high mortality risk DCM patients: follow up time seems to be inadequate to validate a prognostic score, considering the retrospective nature of the study.

Discussion, identification of high mortality DCM patients: can you precise more extensively how a high score could change the management of DCM patient? For example, the clinician should consider a CRT in patients with borderline ECG indications but a high Krakow score? Also the title of the paper “Clinical utility…” claims for it. 

Author Response

  1. Queries about methods:
    1. Population:
  2. As requested more detailed descriptions of derivation and validation cohorts were included in the Methods section: “The derivation cohort comprised of 406 DCM patients (aged 54 ± 14 years, 81% male, NYHA class 2.5 ± 0.9, LVEF 26 ± 9%, left ventricle end-diastolic diameter – 66 ± 10mm, mean NT-proBNP 1476pg/ml) [19].

As for the participating centres, we added the following information on the participating centres from the validation cohort:The validation cohort consisted of 329 consecutive DCM patients from 7 polish cardiac centres from 2009 to 2020 with complete baseline data, 118 (35%) from 2009-2015, and 215 (65%) from 2016-2020; the distinction is motivated by the publication of the European Society of Cardiology (ESC) HF guidelines in 2016.”

  1. We would like to thank the Reviewer for pointing out an important issue, which is the comparison of the derivation and validation cohorts. We fully agree that this is one of the crucial factors in the whole validation process. Therefore, the comparison of the validation and derivation cohorts had been already placed in the Results/Performance of Krakow DCM Risk Score section as well as in Supplementary Table 1 (Table S1) and Supplementary Figure 1 (Figure S1). Moreover, we expanded the subject in the Appendix section.

3.2. Performance of Krakow DCM Risk Score

“Performance of Krakow DCM Risk Score subsection in the Results section: “The validation cohort differed significantly from the derivation cohort in terms of age, HF duration and symptoms, comorbidities (obesity – body mass index, dyslipidemia, anaemia, and chronic kidney disease), heart rate, NT-proBNP, and their required loop diuretics dosage (Table S1, Appendix). However, they did not differ in terms of echocardiographic findings and mortality rates (p=0.97) (Figure S1).

Table S1. Comparison of output and validation cohorts.

Parameters

Derivation cohort

(n=406)

Validation cohort

(n=329)

p-value

Clinical parameters

Age [years]

53.62 ± 13.64

49.88 ± 10.71

<0.0001

Male [n (%)]

330 (81.3%)

264 (79.3%)

0.50

Symptoms’ duration [months]

40.13 ± 58.31

8.84 ± 15.84

<0.0001

BMI [kg/m2]

27.66 ± 5.11

29.41 ± 15.01

0.04

NYHA class

2.51 ± 0.89

2.23 ± 0.77

<0.0001

NYHA III/IV

194 (47.8%)

90 (27%)

<0.0001

Diabetes mellitus [n (%)]

90 (22.2%)

64 (19.2%)

0.33

Prior stroke [n (%)]

24 (5.9%)

13 (3.9%)

0.21

Liver diseases [n (%)]

53 (13%)

47 (14.1%)

0.68

Dyslipidemia [n (%)]

274 (67.5%)

262 (78.7%)

0.0007

COPD [n (%)]

27 (6.7%)

19 (5.7%)

0.60

Atrial fibrillation [n (%)]

129 (31.8%)

101 (30.3%)

0.67

ECG findings

HR [bpm]

81.02 ± 20.3

76.5 ± 16.62

0.007

LBBB [n (%)]

105 (25.9%)

75 (22.5%)

0.29

VT [0/1]

106 (26.2%)

89 (26.6%)

0.89

Echocardiographic findings

LVEF [%]

26.1 ± 9.37

27.02 ± 9.96

0.19

LVEDd [mm]

66.18 ± 10.39

65.08 ± 8.89

0.25

IVS [mm]

10.35 ± 2.18

10 ± 1.98

0.11

LAA [cm2]

29.42 ± 8.42

28.97 ± 8.33

0.45

RAA [cm2]

23.13 ± 8.2

22.2 ± 7.89

0.13

MR moderate/severe [n (%)]

192 (47.3%)

114 (34.2%)

0.0003

TR moderate/severe [n (%)]

105 (25.9%)

66 (19.8%)

0.05

Laboratory results

Hb [g/dl]

14.27 ± 1.59

14.55 ± 1.66

0.003

eGFR [ml/min/1,73m2]

80.01 ± 21.48

83.5 ± 20.9

0.01

NT-proBNP [pg/ml]

3662.8 ± 7616.6

2759.3 ± 3639.6

0.03

LDL [mmol/l]

2.95 ± 0.98

2.99 ± 0.98

0.51

HF therapy

BB [n (%)]

394 (97%)

321 (96.4%)

0.62

ACEi/ARB/ARNI [n (%)]

370 (91.1%)

294 (88.3%)

0.20

MRA [n (%)]

357 (87.9%)

289 (86.8%)

0.64

Furosemide [mg/d]

6.38 ± 19.6

6.26 ± 12.02

<0.0001

Loop diuretics [mg/d]

0.09 ± 0.28

0.16 ± 0.37

<0.0001

CRT [n (%)]

13 (3.2%)

11 (3.3%)

0.93

ICD [n (%)]

39 (9.6%)

30 (9%)

0.78

FU [months]

48.21 ± 31.97

41.63 ± 29.27

0.01

Deaths [n (%)]

70 (17.2%)

50 (14.9%)

0.39

Figure S1. Comparison of mortality rate between derivation and validation cohorts based on Kaplan-Meier estimates.

Appendix:

“The comparison of derivation and validation cohorts.

The validation cohort comprised of younger patients (average age of 3 years less) with shorter and less severe HF symptoms (27% vs. 48% patients with NYHA class III/IV), and lower NT-proBNP level. Despite younger age patients from the validation cohort had a comparable prevalence of comorbidities, including atrial fibrillation and diabetes mellitus. However, they had better kidney function. Surprisingly, though having a shorter HF duration, the validation cohort had no significant differences in terms of echocardiography findings, with similar severity of left ventricle remodelling and systolic dysfunction. Moreover, the implementation of recommended HF therapy was similar. Although the derivation and validation DCM cohorts differed in terms of mentioned parameters, the observed prognosis was similarly poor with the 5-years mortality of over 20%.

Moreover, the summarizing sentences were added to the Krakow DCM Risk Score performance subsection from the Discussion section and are now as follows:

Taking into account the significant differences between the derivation and validation cohorts, including age, HF symptoms and NT-proBNP level, their outcomes were poor with similar 5-years mortality of over 20%. Moreover, the high accuracy of Krakow DCM Risk Score, despite the diversity of DCM cohorts under study, strengthens its value in various DCM cohorts.

  1. End-point definitions:
    1. the primary endpoint:

We believe that all-cause mortality is the most objective measure of outcome. It is independent of any interpretation and is a “zero/one” measurement. We all know that certain causes of death may not be possible to obtain on every occasion, and the reported cause may not be reliable in numerous circumstances, e.g., lack of autopsy, imprecise medical notes, ambiguous ICD coding and finally imprecise information from patients’ family. Moreover, established HF prognostic scales, including GISSI-HF, CHARM, HFSS, MAGGIC, commonly use all-cause mortality as a primary endpoint [Levy WC, Mozaffarian D, Linker DT, et al. The Seattle Heart Failure Model: Prediction of survival in heart failure. Circulation. 2006;113(11):1424–33. Pocock SJ, Ariti CA, McMurray JJV, et al. Predicting survival in heart failure: a risk score based on 39 372 patients from 30 studies. Eur Heart J. 2013;34(19):1404–13. Pocock SJ, Wang D, Pfeffer MA, et al. Predictors of mortality and morbidity in patients with chronic heart failure. Eur Heart J. 2006;27(1):65–75. Barlera S, Tavazzi L, Franzosi MG, et al. Predictors of mortality in 6975 patients with chronic heart failure in the Gruppo Italiano per lo Studio della Streptochinasi nell’Infarto Miocardico-Heart Failure trial. Proposal for a nomogram. Circ Heart Fail. 2013;6(1):31–9].

On the other hand, we agree with the Reviewer opinion that the cause of death is highly relevant in HF and DCM patients. The great majority of death in our cohorts were attributed to CV death (this information had been previously included in the Results section).

  1. the combined endpoint:

We fully agree with the Reviewer that heart transplant (HTX) and left ventricle assistant device implantation (LVAD) are the “hard end-point”, as there is also no room for interpretation. However, the Krakow DCM Risk Score was originally designed for the all-cause mortality prediction [E. Dziewięcka, M. Gliniak, M. Winiarczyk, et al. ESC Heart Failure 2020; 7(5):2455-67]. During the preparation of the Krakow DCM Risk Score, we had performed additional analyses, utilizing a combined endpoint (comprised of death, HTX, LVAD). Patients were stratified according to (1) all-cause mortality (deceased vs. alive) and (2) presence of combined endpoint (death, HTX, LVAD). We had found very similar (almost identical) baseline characteristics of those cohorts (Table A).

Table A. Results of the comparisons between dead and alive patients or patients with and without composite endpoint (death, LVAD implantation, HTX).

p-value

(deceased vs. alive)

p-value

(with vs. without composite endpoint)

Age [year]

0.34

0.37

Male [n (%)]

0.29

0.48

BMI [kg/m2]

0.07

0.12

HF hospitalisation [n (%)]

0.02

0.004

Blood pressure [mmHg]

Systolic

Diastolic

0.01

0.03

0.01

0.005

Etiology [n (%)]

Inflammatory

Toxic

Tachyarythmic

Familial

Other

Unknown

0.69

0.84

Symptoms’ duration [month]

0.002

0.002

Diabetes mellitus [n (%)]

0.07

0.21

CV prior hospitalisation [n (%)]

6 months

12 months

>12 months

0.56

0.73

0.0006

0.62

0.36

0.12

Current smoker [n (%)]

0.87

0.94

COPD [n (%)]

0.73

0.88

AF [n (%)]

0.42

0.53

Dementia [n (%)]

0.17

0.72

Neoplasm [n (%)]

0.25

0.47

Cerebrovascular disease [n (%)]

0.0001

0.11

Dyslipidaemia [n (%)]

0.002

0.01

Liver disease [n (%)]

0.13

0.38

Oedema [n (%)]

0.04

0.05

Killip class

0.05

0.34

NYHA class

<0.0001

<0.0001

HR [bpm]

0.48

0.54

QRS [ms]

0.31

0.34

Intraventricular delay [n (%)]

0.69

0.26

EF [%]

0.01

0.009

LVEDd [mmm2]

<0.0001

0.02

LAd [mm]

0.02

0.003

RVd [mm]

0.08

0.10

TAPSE [mm]

0.59

0.85

LAA [cm2]

0.04

0.05

RAA [cm2]

0.03

0.03

E wave [m/s]

0.02

0.02

PASP [mmHg]

0.0001

0.002

Mild or severe MR [n (%)]

0.07

0.12

Mild or severe TR [n (%)]

0.04

0.04

nsVT [n (%)]

0.046

0.03

WBC [K/ul]

0.38

0.61

Lymphocytes [%]

0.0008

0.0008

Hb [g/dl]

0.0003

0.0003

Anaemia [n (%)]

0.006

0.09

Creatinine [umol/l]

GFR>60ml/min [n (%)]

GFR 60-30ml/min [n (%)]

GFR<30ml/min [n (%)]

0.11

0.008

0.02

0.28

0.11

0.08

0.03

0.74

Na [mmol/l]

0.07

0.07

K [mmol/l]

0.53

0.61

AlAT [U/l]

0.07

0.07

AspAT [U/l]

0.85

0.86

Glucose [mg/dl]

0.56

0.56

Cholesterol [mmol/l]

0.0005

0.0005

Cholesterol LDL [mmol/l]

0.003

0.003

TSH [uIU/ml]

0.69

0.85

hsTnT [ng/ml]

<0.0001

<0.0001

CRP [mg/l]

0.0007

0.0007

log10 of NT-proBNP

<0.0001

<0.0001

ACEi/ARB/ARNI [n (%)]

0.03

0.02

Beta-blocker [n (%)]

0.62

0.0006

MRA [n (%)]

0.32

0.53

Furosemide [mg/day]

0.007

0.001

Torasemide [mg/day]

0.20

0.39

Digoxin [n (%)]

0.004

0.004

Statins [n (%)]

0.06

0.002

ICD [n (%)]

0.15

0.28

CRT/CRT-D [n (%)]

0.04

0.0001

  1. Queries about results:
    1. Baseline characteristics:
      1. For more clarity of the presented results, we change the first sentence from the Baseline characteristics subsection as follows: “During a follow-up of mean 41.6 ± 29.3 months, 49 (15%) patients of validation cohort died: 41 (85%) stemming from cardiovascular causes (38 patients – due to HF worsening, 3 patients – SCD), and 8 patients died as a result of neoplasms”.
      2. As suggested, we added the column “All” to Table 1 (as presented below).

Table 1. Baseline characteristics.

Parameters

All

n=329

Alive

n=280 (85%)

Deceased

n=49 (15%)

p-value

Clinical parameters

Age [years]

49.88 ± 10.71

49.31 ± 10.69

53.29 ± 10.66

0.006

Male [n (%)]

261 (79.3%)

226 (80.7%)

35 (71.4%)

0.14

Symptoms’ duration [months]

21.3 ± 34.78

25.15 ± 89.865

28.63 ± 41.03

0.31

BMI [kg/m2]

29.39 ± 14.37

29.47 ± 16.162

29.16 ± 6.18

0.77

NYHA class

2.23 ± 0.77

2.14 ± 0.73

2.73 ± 0.8

<0.0001

NYHA III/IV

89 (27.1%)

63 (22.5%)

26 (53.1%)

<0.0001

Diabetes mellitus [n (%)]

64 (19.5%)

49 (17.5%)

15 (30.6%)

0.03

Prior stroke [n (%)]

13 (4%)

11 (3.9%)

2 (4.1%)

0.96

Liver diseases [n (%)]

47 (14.3%)

39 (13.9%)

8 (16.3%)

0.66

Dyslipidemia [n (%)]

258 (78.4%)

223 (79.6%)

35 (71.4%)

0.20

COPD [n (%)]

18 (5.5%)

15 (5.4%)

3 (6.1%)

0.83

Atrial fibrillation [n (%)]

100 (30.4%)

84 (30.0%)

16 (32.7%)

0.71

Hypertension [n (%)]

155 (47.1%)

127 (45.4%)

28 (57.1%)

0.13

SBP [mmHg]

122.99 ± 20.55

123.97 ± 20.58

117.83 ± 19.94

0.09

ECG findings

HR [bpm]

76.5 ± 16.62

75.76 ± 16.011

80.93 ± 19.75

0.07

QRS [ms]

105.59 ± 35.24

104.11 ± 34.077

114.49 ± 41.13

0.05

LBBB [n (%)]

74 (22.5%)

62 (22.1%)

12 (24.5%)

0.72

VT [0/1]

88 (26.7%)

74 (26.6%)

14 (28.6%)

0.81

Echocardiographic findings

LVEF [%]

27.02 ± 9.96

27.73 ± 9.936

23.72 ± 9.55

0.01

LVEDd [mm]

65.08 ± 8.89

64.76 ± 8.64

66.46 ± 10.24

0.24

IVS [mm]

10 ± 1.98

10.06 ± 1.987

9.7 ± 1.91

0.33

RVOT [mm]

33.79 ± 6.61

33.4 ± 6.383

36.15 ± 7.41

0.008

TAPSE [mm]

19.14 ± 4.12

19.41 ± 4.118

17.77 ± 4.02

0.01

LAA [cm2]

28.97 ± 8.33

28.35 ± 7.813

32.04 ± 10.27

0.01

RVSP [mmHg]

25.46 ± 13.12

23.89 ± 11.886

33.46 ± 16.15

0.0001

MR moderate/severe [n (%)]

111 (33.7%)

89 (31.8%)

22 (44.9%)

0.07

TR moderate/severe [n (%)]

66 (20.1%)

44 (15.7%)

22 (44.9%)

<0.0001

Laboratory tests results

Hb [g/dl]

14.55 ± 1.66

14.67 ± 1.626

13.88 ± 1.78

0.002

eGFR [ml/min/1,73m2]

83.5 ± 20.9

84.91 ± 20.411

75.67 ± 22.71

0.006

NT-proBNP [pg/ml]

2759.25 ± 3639.6

2297.9 ± 3131.6

4980.7 ± 4910.9

<0.0001

LDL [mmol/l]

2.99 ± 0.98

2.99 ± 0.969

2.86 ± 1.04

0.36

Heart failure therapy

BB [n (%)]

317 (96.4%)

272 (97.1%)

45 (91.8%)

0.049

ACEi/ARB/ARNI [n (%)]

291 (88.4%)

253 (90.4%)

38 (77.6%)

0.01

MRA [n (%)]

285 (86.6%)

244 (87.1%)

41 (83.7%)

0.51

Loop diuretics [mg/d] 1

44.47 ± 69.24

37.91 ± 56.42

80.16 ± 113.94

0.0003

Furosemide [mg/d]

25.7 ± 50.31

22.02 ± 44.717

45.57 ± 72

0.03

Ivabradine [n (%)]

53 (16.1%)

41 (14.6%)

12 (24.5%)

0.08

Digoxin [n (%)]

52 (15.8%)

38 (13.6%)

14 (28.6%)

0.008

Statins [n (%)]

148 (45%)

124 (44.3%)

24 (49.0%)

0.54

CRT [n (%)]

11 (3.3%)

6 (2.1%)

5 (10.2%)

0.004

ICD [n (%)]

30 (9.1%)

23 (8.2%)

7 (14.3%)

0.17

1Loop diuretics dosages were calculated as the sum of the furosemide daily dosage, and 3 times the torsemide daily dosage.

Abbreviations: BMI – body mass index, NYHA – New York Heart Association class, COPD – chronic obstructive pulmonary disease, SBP – systolic blood pressure, HR – heart rate, LBBB – left bundle branch block, VT – ventricular tachyarrhythmia, LVEF – left ventricle ejection fraction, LVEDd – left ventricle end-diastolic diameter, obtained from parasternal long-axis view (PLAX), IVS – intraventricular septum thickness obtained from PLAX, RVOT – right ventricle outflow track diameter obtained from PLAX, MR/TR – mitral/tricuspid regurgitation, Hb – haemoglobin, eGFR – estimated glomerular filtration rate, NT-proBNP – N-terminal prohormone B-type natriuretic peptide, LDL – low-density lipoprotein, BB – beta-blocker, ACEI – angiotensin-converting-enzyme inhibitor, ARB – angiotensin receptor blocker, ARNI – angiotensin receptor—neprilysin inhibitor, MRA – mineralocorticoid receptor antagonist, ICD – implantable cardioverter-defibrillator, CRT – cardiac resynchronization therapy.

  • Moreover, as suggested we change the term “non-survivors” into “deceased patients”, as follows and “survivals” into “alive”.

  1. High mortality risk DCM patients:
    1. The mean time of observation of the study population was 42 ± 29 months (3.5 years). As presented in the supplemented files in the “derivation study” most prognostic scales used in the HF population was derived from the HF cohorts with an observational period between 3 and 4 years (Table B) [E. Dziewięcka, M. Gliniak, M. Winiarczyk, et al. ESC Heart Failure 2020; 7(5):2455-67].

Table B. Comparison of derivation cohort from HF prognostic scales.

BCN Bio-HF

CHARM

EMPHASIS

GISSI-HF

MAGGIC

MUSIC

OPTIMIZE-HF

SHFM

Miura et al. score

Inclusion and observation period

2006-2010

1999-2003

2006-2010

2002-2005

1980s-2008

2003-2007

2003-2004

1992-1994

1994-1999

Follow-up time [years]

3.4

3.2

2.1

3.9

2.5

3.7

4 days

1.2

5

  1. Queries about the discussion - identification of high mortality DCM patients:
    1. As suggested, we expanded the Identification of high mortality DCM patients subsection from the Discussion section and is now as follow:

Accurate prediction of long-term outcomes, including mortality is a cornerstone of comprehensive HF management. Although the overall prognosis in DCM is poor, patients’ individual prognoses may be highly variable. Therefore, the development of an accurate prognostic risk model has the potential for more comprehensive and tailored management. Quantifying patients’ survival predictions based on their overall risk profile can help identify those in need of more concentrated monitoring and more intensive HF therapy. Additional potential use of the model includes educating patients on the significant value of HF medications. When altering their implementation in the online calculator, patients can be presented with a higher mortality risk without proper pharmacotherapy. Moreover, patients with high mortality risk should be earlier listed for cardiac transplantation or counselled about end-of-life issues.

As stated above, the identification of patients with high mortality risk is crucial in everyday practice in HF and DCM patients. We recognized the high mortality risk as a 2-year mortality risk estimated above 6.0% (75th percentile). The accuracy of this cut-off point is high, and patients with a calculated risk of > 6.0% had a 5-times higher mortality risk during follow-up.”

Reviewer 2 Report

In this paper, the authors aimed to validate the Krakow DCM Risk Score (previously proposed by the same team) and to establish a cut-off point for high-risk patients. Dilated cardiomyopathy (DCM) is an important cause of heart failure (HF). Although there are a number of prognostic models currently used for pts with HF, dedicated tools for DCM are not established.

The study population consisted of 735 patients, with 329 pts in the validation cohort (followed up for a mean of 42 months). The authors found that the score had good predictive accuracy. A 2-year mortality risk of > 6.0% identified high-risk pts. 

The study is well designed, written, and presented. The findings of the study are clinically relevant.

I do not have specific concerns regarding the paper.

Author Response

We would like to thank the Reviewer for the time and effort spend in order to carefully review our manuscript. We are very pleased to learn that it was found to be of interest.